# Changes in Canadian Adolescent Well-Being since the COVID-19 Pandemic: The Role of Prior Child Maltreatment

**DOI:** 10.3390/ijerph191610172

**Published:** 2022-08-17

**Authors:** Jacinthe Dion, Catherine Hamel, Camille Clermont, Marie-Ève Blackburn, Martine Hébert, Linda Paquette, Daniel Lalande, Sophie Bergeron

**Affiliations:** 1Département des Sciences de la Santé, Université du Québec à Chicoutimi, 555 bl. Université, Chicoutimi, QC G7H 2B1, Canada; 2ÉCOBES-Recherche et Transfert, Cégep de Jonquière, Pavillon Manicouagan, 3791 de la Fabrique St., Jonquière, QC G7X 7W2, Canada; 3Département de Sexologie, Université du Québec à Montréal, C.P. 8888, Succursale Centre-Ville, Montréal, QC H2L 2C4, Canada; 4Département de Psychologie, Université de Montréal, C.P. 6128, Succursale Centre-Ville, Montréal, QC H3C 3J7, Canada

**Keywords:** COVID-19 stress, adolescents, psychological adaptation, child abuse and neglect, anxiety, depression, conduct disorder, self-esteem, life satisfaction, longitudinal design

## Abstract

Adolescents may be particularly vulnerable to the negative impact of the coronavirus disease 2019 (COVID-19) pandemic, given their increased socialization needs during this developmental period. This prospective study examined the potential changes in adolescents’ well-being from before to during the pandemic, and the moderating role of a history of child maltreatment (CM), COVID-19-related distress, and gender among 1,802 adolescents (55.5% participants identified as boy, 42.2% as girl, and 1.5% as nonbinary; M_age_ 14.74 years). Another aim was to determine whether COVID-19-related distress mediated the relationship between CM and well-being. Results revealed that COVID-19-related distress was associated with lower well-being (i.e., higher levels of internalized and externalized behaviors, and lower levels of self-esteem and life satisfaction). Boys experienced a greater decrease in life satisfaction and self-esteem than girls. A history of CM had a moderation effect, with the pandemic having a lesser impact on the outcomes of adolescents with such a history. However, it was also associated with more COVID-19-related distress, which in turn was associated with lower levels of internalized and externalized behaviors, self-esteem, and life satisfaction. These unexpected results with regard to CM might indicate that the social restrictions during the pandemic could have had a relieving effect on adolescents with particular challenges associated with CM.

## 1. Introduction

In response to the ongoing coronavirus disease 2019 (COVID-19) pandemic, the Canadian government implemented several restrictive measures to decrease physical contact and increase physical distancing. During the spring of 2020, schools, museums, theaters, and many other social, recreational, and cultural centers were closed, and mass gatherings and events were canceled. One of the consequences of these measures is that adolescents were largely prevented from meeting their friends [1]. Although schools subsequently reopened for the 2020–2021 school year, significant restrictions remained, with some schools temporarily shutting down, in addition to the shift in education towards the distance-learning mode when the number of infections became too high. Furthermore, extracurricular activities were either quite limited or not offered at all during the 2020–2021 school year.

Although these measures were necessary to limit the transmission of the virus, there are reasons to be concerned about their mental health impact, especially among adolescents, who are the group with the greatest need to socialize with their peers [2]. Adolescence is an important life stage during which key developmental experiences occur, such as a first romantic relationship, self-assertion, school involvement, and professional development, all of which a pandemic may impede. Likewise, the negative effects of pandemic restrictions and related stressors may be heightened among this vulnerable age group [3,4] because adolescence is a time of transition toward adulthood with an increased need for autonomy and greater reliance on peers and intimate relationships for social support [2].

It is well documented that in previous epidemics (e.g., severe acute respiratory syndrome (SARS), Ebola virus disease, and influenza A (H1N1) virus), the periods of quarantine and isolation in adolescents were associated with stressors such as prolonged duration of social distancing, fears of infection, frustration and boredom, inadequate information about the virus, forced physical distance from friends and intimate partners, and lack of personal space at home [5]. The unique and unprecedented crisis of COVID-19 shares certain similarities: the health risks and related restrictions have negatively impacted individuals and their communities, particularly adolescents, triggering various mental health problems, including anxiety, depression, and lower life satisfaction [6,7,8,9]. During the pandemic, the prevalence of these issues seems to be higher among adolescents than among the general population [3].

A recent longitudinal study also suggests that the COVID-19 pandemic has contributed to the onset of mental health problems [10] or amplified pre-existing symptoms in adolescents [11]. However, these data were collected during the first months of the pandemic, precluding any understanding of their well-being a year after its onset. Moreover, only a few longitudinal studies included baseline measures before the COVID-19 pandemic began, yet these data are needed to better understand factors that may relate to changes in mental health symptoms over time. For instance, a study of 248 adolescents in Australia found significant increases, but of low magnitude, in adolescents’ depressive symptoms and anxiety and a significant decrease in life satisfaction [6]. These changes were more pronounced among adolescent girls versus boys and among those experiencing greater COVID-19-related distress.

In addition, the potential effects of pandemic restrictions and related distress may be heightened among adolescents who have experienced prior adverse childhood experiences, such as child maltreatment (CM), because these experiences may increase vulnerability to future life stressors [12,13,14]. According to the World Health Organization, CM “includes all types of physical and/or emotional ill-treatment, sexual abuse, neglect, negligence and commercial or other exploitation, which results in actual or potential harm to the child’s health, survival, development or dignity in the context of a relationship of responsibility, trust or power” [15].

Independent of the pandemic, those who have experienced CM tend to also experience greater rates of stigma, social isolation, and mental health issues compared with their non-victimized peers [16,17]. Although previous correlational findings indicate that adolescents who experienced CM may experience more negative effects from the pandemic [18], there are significant gaps concerning the extent to which belonging to this high-risk group prospectively impacts different aspects of mental health over time in the aftermath of a triggering event as far-reaching as the COVID-19 pandemic. It is likely that a history of CM may worsen psychological symptoms. It is also possible that it leads to more distress experienced during the pandemic, which in turn leads to more psychological symptoms. Therefore, to better understand the well-being among maltreated adolescents, it is important to investigate whether (1) CM is a risk factor (moderator) that may increase the deterioration observed among adolescents during the pandemic, and whether (2) COVID-19-related distress is a key process (mediator) that transmits the detrimental effect of CM.

Against this backdrop, the first aim of this prospective study was to examine changes in Canadian adolescent well-being, from before the onset of the pandemic (*pre-pandemic*, hereafter *T1*) to 1 year after the pandemic began (*during the pandemic*, hereafter *T2*). Although there is no standard to define well-being, and multiple indicators have usually been used across studies [19], in this study, four measures were used to assess both deficit (i.e., internalized and externalized behaviors) and positive indicators (i.e., self-esteem and life satisfaction). The study prediction was that from T1 to T2, adolescents would report an increase in internalized and externalized behaviors, and a decrease in self-esteem and overall life satisfaction. The second aim of the study was to examine the moderating roles of CM and of COVID-19-related distress in the changes in outcomes between T1 and T2. It was anticipated that adolescents with a history of CM and those with higher levels of COVID-19-related distress would experience lower levels of well-being. The third aim of the study was to determine if COVID-19-related distress mediated the links between CM and the four outcomes. Finally, given the recent literature indicating that girls may endorse more pronounced mental health symptoms than their male peers [6], another aim of this study was to test whether gender differences were evident in the changes between T1 and T2, and whether it moderated the mediational models. Findings may increase our understanding of the possible consequences of the social/physical distancing measures in the ongoing COVID-19 pandemic, as well as in future pandemics—predicted to be more frequent in the coming years [20]. Importantly, these results will inform the development of targeted interventions to better help adolescents, particularly high-risk subgroups (i.e., adolescents who have experienced CM and those experiencing more distress related to the pandemic).

## 2. Materials and Methods

### 2.1. Participants

The baseline sample (T1) comprised 1802 adolescents aged 14–18 years (mean 14.74 years; *SD* = 0.84). Of the whole sample, most participants identified at T1 with the Quebecois culture (67.2%) and 32.8% reported other cultural identities, including Canadian, American, East European, West European, African, Asian, Middle Eastern, Latin/South American, and Caribbean cultures. Most adolescents (64.4%) lived with both parents during the pandemic. Regarding gender identity, 55.5% participants identified as boy, 42.2% as girl, 1.5% as nonbinary, gender fluid, two-spirit, or “other”, and 0.9% of participants refused to answer.

The follow-up sample (T2) comprised 825 adolescents aged 15–19 years (mean 15.84 years; *SD* = 0.73), representing 45.4% of the baseline sample. Among those participants, 51.9% identified as girl, 46.5% as boy, and 1.5% as nonbinary, gender fluid, or others. Attrition analyses revealed that gender (being a boy) was associated with greater dropouts at T2. Although no significant differences were observed for internalized behaviors and self-esteem, small differences were found for the two other outcomes. Specifically, slightly higher levels of externalized behaviors and slightly lower levels of life satisfaction were associated with dropouts at T2 (η^2^ = 0.005 and 0.007, respectively).

### 2.2. Procedure

Data were collected as part of an ongoing Canadian longitudinal study concerning sport participation and resilience. Ensuring sample diversity, this study recruited adolescents in the ninth or tenth grade (at T1) in Canadian schools from urban, semi-urban, and rural areas and from different socioeconomic backgrounds. Participants completed a self-report, anonymous survey (Qualtrics Research Suite) in their classrooms on electronic tablets provided by research assistants in the first wave (T1, between October and December 2019) and second wave (T2, between November 2020 and June 2021) of this study. For the second wave, our research team modified the study procedure to follow the recommended COVID-19 safety measures as follows: a medical form needed to be completed by research assistants before data collection; during data collection, they used face covers and washed their hands often. Moreover, they disinfected all materials between data collection in different classes (e.g., electronic tablets).

Before enrollment in both study waves, participants received detailed information about the study and provided informed consent. Of note, in Quebec, adolescents aged 14 and older can provide their own informed consent. The questionnaires were both completed in an average of 40 min at each wave, including three attention-testing questions. Study participation was compensated for with a reusable water bottle and draws (lottery gift certificates). A code was used as an anonymous identifier. To generate their identification code, participants answered eight questions (e.g., the first letter of your mother’s or female caregiver’s first name, the first letter of the city where they were born). The research procedure was approved by the first author’s Institutional Review Board and was performed according to the Declaration of Helsinki.

Of the six schools that were approached to participate in this study, five agreed to participate. At T2, one of the five schools refused to participate in the study due to the COVID-19 pandemic, which represents 22.2% of the sample at T1 (*n* = 412). Analyses indicated no significant differences in the variables of interest between the students of that school and others. Therefore, we decided to retain the data from all schools in the analyses.

At both T1 and T2, more than 98% of the adolescents who were approached agreed to participate in this study. Nonetheless, at T1, 36 participants were excluded because they did not provide a correct answer to all of the three attention-testing questions or because of inconsistent response patterns. At T2, of the 1069 students who agreed to participate, the T1 and T2 data for 233 participants could not be matched (because of non-matching identifiers linking T1 and T2), and 11 participants were excluded because they failed all of the three attention-testing questions, resulting in a final sample of 825 participants in T2.

### 2.3. Measures

Participants provided **demographic information,** including their age, culture, the language of origin, and gender identity.

Several behavior-specific questions were used to assess **child maltreatment** experienced before 14 years of age. This approach was in accordance with the recommendation by Stoltenborgh et al. to obtain a better estimate of CM [21]. Thus, five forms of CM were assessed: *sexual*, *emotional*, and *physical abuse*, *neglect*, and *exposure to domestic violence*. First, sexual abuse was measured using three dichotomous items (yes/no) from a subscale of the Early Trauma Inventory Self-Report–Short Form (ETISR-SF) [22], which was translated in French and adapted in a Canadian survey with a large sample size [23]. Second, two adapted items of the ETISR-SF were used to assess emotional abuse during childhood [22,23]. Third, physical abuse was measured by one item from the Longitudinal Study of Adolescent Health [24]. Fourth, two items from the Neglect subscale of the Intimate Partner Violence questionnaire were used to measure parental neglect [24]. Finally, exposure to domestic violence was assessed by three items adapted from the Revised Conflict Tactics Scale [25]. Lastly, all indicators of adversity were dichotomized (i.e., 0 = *participant did not experience the trauma*; 1 = *participant experienced the trauma at least once or more frequently*) to compute a total cumulative childhood maltreatment score varying between 0 (*no maltreatment*) and 5 (experienced five *forms of maltreatment*).

**Distress related to the COVID-19 pandemic** was measured using an adapted French version, suitable for adolescents, of the COVID-19 Peritraumatic Distress Index (CPDI), originally developed and validated for the Chinese population and translated in English [26], but further validated in various languages such as Italian, Spanish, and Odia (India) [27,28,29]. The 24-item CPDI assesses the frequency of anxiety, depression, specific phobias, cognitive changes, avoidance and compulsive behaviors, physical symptoms, and loss of social functioning on a scale ranging from 0 (*never*) to 4 (*most of the time*). The total raw score ranged from 0 to 96, with a higher score indicating a higher level of distress. This questionnaire showed a high internal consistency (α = 0.90).

**Internalizing and externalizing behavior problems** were assessed using the French validated version [30,31] of the *emotional symptoms* and *conduct problems* subscales of the Strengths and Difficulties Questionnaire (SDQ), respectively [32]. Internalized behaviors are defined as behaviors directed inwardly toward the self [33]. In this study, using the *emotional symptoms* subscale of the SDQ, internalized behaviors included anxiety and depression symptoms. Externalized behaviors are directed outwardly toward the social environment [34], and the *conduct problems subscale* assessed aggression and conduct problems, such as lying, bullying, and stealing. Both subscales comprise 10 items (5 for internalized behaviors and 5 for externalized behaviors), for which the participants answered based on a 3-point Likert-type scale (1 = *not true*, 2 = *a little true*, 3 = *true*). A higher score on the scale of internalized or externalized behaviors indicated a greater frequency of adoption of these behaviors. The internalized behavior subscale demonstrated adequate consistency for T1 and T2 (α = 0.81 and 0.83, respectively). However, it was quite low for externalized behaviors—an item (the reverse score of “I usually do as I am told”) had to be removed from the original subscale to increase consistency (α = 0.58 and 0.49 for T1 and T2, respectively).

A 4-item version of the Self-Description Questionnaire [35], derived from the National Longitudinal Survey of Children and Youth (NLSCY) [36], was used to measure adolescents’ **self-esteem**, that is, confidence in personal worth and value. The participants answered each item on a scale ranging from 0 (*false*) to 4 (*true*), and the total score ranged from 0 to 16. The scale demonstrated high internal consistency in T1 and T2 (α = 0.85 and 0.88, respectively).

**Life satisfaction** was assessed using the 5-item Satisfaction with Life Scale (SWLS) [37], also validated among French adolescents [38], which measured how satisfied participants were with their lives. They were asked to respond on a 7-point Likert-type scale ranging from 1 (*strongly disagree*) to 7 (*strongly agree*). The estimated internal consistencies were α = 0.86 for T1 and 0.87 for T2.

### 2.4. Statistical Analyses

Descriptive statistics were obtained through SPSS version 28.0 (IBM Corp: Armonk, NY, USA) [39]. Paired sample *t*-tests were conducted to determine whether the levels of internalizing and externalizing behaviors, self-esteem, and life satisfaction significantly increased or decreased between T1 and T2. Missing data at the variable level were minimal; mean scores were used where possible to reduce the impact of missing data.

Potential moderators of change in the scores for well-being between T1 and T2 (i.e., CM and COVID-19-related distress) were assessed using Model 2 of the MEMORE code in Mplus [40] to test and probe moderation in two-instance repeated measures designs. This approach computes a pre–post difference score and determines if the potential moderator predicts the difference [41]. Significant interactions were centered at the 16th, 50th, and 84th percentiles for COVID-19-related distress and at values of 0, 2, and 3 for CM.

In addition, four hypothesized mediational models were tested using path analyses with Mplus 8 [42]. Specifically, the study examined whether COVID-19-related distress mediated the associations between CM and the four outcomes (internalizing behaviors, externalizing behaviors, self-esteem, and life satisfaction) at T2. Outcomes at T1 were entered as control variables. The bias-corrected bootstrap method (10,000 iterations) at 95% confidence interval (CI) was used to evaluate the indirect effects of cumulative CM [43,44]. A multiple-group gender invariance path analysis was conducted using a corrected chi-squared difference test (Satorra–Bentler scaled chi-square) to evaluate the gender moderation hypothesis for the mediational models: a significant chi-squared difference between the configural and the constrained models indicated the existence of differences between boys and girls. The mediational models were first estimated using path analyses and gender was then examined as a potential moderator. The small subsample of individuals who self-identified as nonbinary (*n* = 26 at T1 and *n* = 5 at T2) rendered it impossible to estimate the moderation and mediational models for such respondents. Nevertheless, we provided descriptive statistics for them.

For both types of analyses in Mplus, the models were fully saturated, as the associations between all variables were estimated (χ^2^ = 0; *df* = 0, comparative fit index (CFI) = 1.00; Tucker–Lewis index (TLI) = 1.00; root-mean-square error of approximation (RMSEA) = 0.00). These models were also tested using the maximum likelihood (ML) estimator, and missing data were handled using the full information maximum likelihood (FIML) estimation method [42]. FIML is comparable to multiple imputation [45]; however, instead of replacing the missing data points, it uses partially available information from one case to adjust parameter estimates with missing data and gives larger weights to individuals with more repeated assessments [46].

## 3. Results

### 3.1. Demographics and Descriptive Statistics

Table 1 presents the descriptive statistics of study variables. Correlations between these variables are presented in Table 2. The variables were significantly correlated within and between time points in the predicted direction mostly in line with the proposed hypotheses. Paired sample *t*-tests (T1–T2) of each outcome are shown in Table 3. Results revealed small but significant changes between T1 and T2 for self-esteem and life satisfaction, that is, participants reported lower levels of self-esteem, life satisfaction, and externalized behaviors at T2. However, no significant changes were observed regarding internalized behaviors, and a decrease was observed in externalizing behaviors. Finally, on average, girls reported significantly more CM (*p* < 0.001), more symptoms of internalized behaviors (*p* < 0.001) and lower levels of self-esteem (*p* < 0.001) and life satisfaction (*p* = 0.026) at both T1 and T2 than boys. Non-binary adolescents also reported significantly more COVID-related distress, more internalized and externalized behaviors at T1, and lower self-esteem and life satisfaction at T1 and T2 (all *ps* < 0.005) than cisgender adolescents. However, these preliminary results should be interpreted with caution, given low statistical power.

### 3.2. Moderators of Changes in Well-Being from T1 to T2

Results of the moderation models are displayed in Table 4. Results of the mediation models are shown in Table 5.

#### 3.2.1. Child Maltreatment

CM was a significant moderator of all outcomes, that is, having experienced two or more forms of CM was associated with a decrease in internalized and externalized behaviors from T1 to T2. Moreover, adolescents who reported having experienced three or more forms of CM also reported a decrease in externalized behaviors and an increase in self-esteem. Finally, adolescents who reported fewer than three forms also reported a decrease in life satisfaction, whereas having experienced three or more forms of CM was associated with no changes in life satisfaction.

#### 3.2.2. COVID-19-Related Distress

The results of the present study indicated that COVID-19-related distress moderated changes in the four outcomes at T2 (internalizing behaviors, externalizing behaviors, self-esteem, and life satisfaction). Specifically, high levels of distress were associated with a greater increase in internalized behaviors, and moderate to high levels of distress with greater decreases in self-esteem and life satisfaction. Low to moderate levels of COVID-19-related distress were associated with a decrease in externalized behaviors. For adolescents with higher levels of COVID-19-related distress, no significant change was observed in externalized behaviors.

#### 3.2.3. Gender

Although girls reported higher levels for all outcomes at T1 and T2, except for externalized symptoms, participant gender did not moderate the changes in internalized and externalized behaviors. However, it significantly moderated the change scores in self-esteem and life satisfaction. Specifically, boys reported a decrease in self-esteem and life satisfaction but girls did not.

### 3.3. COVID-19-Related Distress as a Mediator of Associations between CM and Outcomes

To better understand the impact of CM on well-being among adolescents, path analyses were conducted with COVID-19-related distress as a mediator for each outcome at T2 (see Table 4). While controlling for each outcome at T1, the findings indicated that prior CM, measured at T1, was positively and significantly associated with COVID-19-related distress measured at T2. In turn, COVID-19-related distress was significantly associated with higher levels of internalized and externalized behaviors, as well as lower levels of self-esteem and life satisfaction at T2. Results also indicated that CM was associated with higher levels of internalized and externalized behaviors and lower levels of self-esteem and life satisfaction at T1 and T2. Results of the gender invariance analyses indicated that the models did not differ between girls and boys for all outcomes.

## 4. Discussion

This prospective study investigated the changes in well-being among adolescents from the beginning of the COVID-19 pandemic, including assessment of three potential moderators: a history of CM, COVID-19-related distress, and gender. This study also examined the mediating role of COVID-19 distress in the associations between CM and outcomes. Overall, the adolescents appeared to cope well during the pandemic as only slight decreases were observed for two of the four indicators of well-being. However, the results also revealed that the adolescents who reported more distress related to the pandemic also experienced a decrease in well-being: more internalized and externalized symptoms and less self-esteem and life satisfaction. Boys also reported a decrease in self-esteem and life satisfaction over time. Finally, although CM was associated with an increase in well-being from T1 to T2, those who experienced CM had worse well-being than their non-victimized counterparts, via COVID-19 related distress.

### 4.1. Changes in Well-Being from Pre-to during COVID-19 Pandemic

As expected, the results suggest that adolescents’ self-esteem and life satisfaction have slightly deteriorated from the pre-pandemic levels. While this is the first study to have investigated changes in self-esteem, the results concerning life satisfaction are consistent with prior work [6]. Contrary to the study hypotheses, no changes were observed in internalized behaviors; however, a decrease in externalized behaviors was observed. These results are less consistent with those of another study [5] and a review of longitudinal studies [47], which concluded that depressive symptoms increased at the beginning of the pandemic among adolescents. Hence, the results of a meta-analysis of longitudinal studies also found that the psychological impact of COVID-19 pandemic lockdowns was rather small and varied greatly among individuals, suggesting resilience in the population [48]. Our results suggest that a year after the beginning of the pandemic, the decrease in well-being may not be as detrimental as it was at its beginning or may fluctuate according to the level of distress experienced by adolescents.

### 4.2. COVID-19-Related Distress and Child Maltreatment

The current findings also expand prior research by identifying how changes in mental health well-being—specifically internalized and externalized behaviors, self-esteem, and life satisfaction—during the pandemic may vary based on the moderating roles of COVID-19-related distress and a history of CM. Our results indicate that higher levels of COVID-19-related distress acted as a moderator of changes observed between the pre-pandemic and intra-pandemic periods, including increases in internalized and externalized behaviors and decreases in self-esteem and life satisfaction. These results agree with those of studies conducted in the beginning of the pandemic involving a smaller sample of adolescents [6] and college students [49], which reported the moderating role of stress related to the pandemic in increasing depressive symptoms and anxiety and decreasing life satisfaction.

However, contrary to our hypothesis, the study results suggest that the well-being of adolescents who reported experiencing CM did not deteriorate during the pandemic. Although it may be suggested that in the context of the pandemic and related stressful and traumatic events, adolescents with a history of CM may be more likely to develop mental health issues that could affect their development; the opposite may be true instead—those with a CM history did not experience a decline in their well-being, which may be attributable to the social restrictions that could have had a relieving effect on adolescents with particular challenges associated with maltreatment. It may also be because they are adapted to higher levels of stress. For example, the results of a study by Hamza et al. (2021) revealed that although postsecondary students with pre-existing mental health concerns continued to present mental health concerns during the pandemic, they showed similar or improving mental health during the pandemic compared with the previous (pre-pandemic) year [50]. Another study conducted in the Netherlands also reported decreases in adolescent mental health symptoms among those who had the highest clinical severity of symptoms pre-pandemic [51]. Taken together, these results may suggest that more vulnerable adolescents develop resilience in the face of other adversities, such as the COVID-19 pandemic.

In contrast, adolescents who reported CM also presented higher levels of internalized and externalized behaviors as well as lower levels of self-esteem and life satisfaction at both T1 and T2, compared with adolescents who did not report CM. Our mediational analyses provided additional understanding of the role of CM in adolescent adaptation through the pandemic, that is, results of the four mediation models indicated that CM was associated with higher levels of COVID-19-related stress, which in turn was associated with higher levels of internalized and externalized behaviors and lower levels of self-esteem and life satisfaction. Therefore, although our results suggest that adolescents with a history of CM may appear to experience improvement during the pandemic, we should be careful with our interpretation as these adolescents may be more vulnerable to experiencing distress when facing other stressors, which in turn decreases their overall well-being. To the best of our knowledge, this study is the first to prospectively investigate the impact of CM among adolescents on changes during the pandemic; other work is needed to better understand how these more vulnerable youth cope with distress induced by the COVID-19 pandemic.

### 4.3. Gender Differences

The study findings regarding gender were surprising: it was not a moderator of change scores in internalized and externalized behaviors; however, boys—and not girls—experienced declines in self-esteem and life satisfaction adaptation. These results are contrary to those of other studies that have shown that girls experienced greater declines than boys [6,49]. Our study results may reflect the lower level of well-being on the four outcomes observed pre-pandemic, which means that well-being did not decrease more because it was already low. These lower levels are also consistent with the well-established literature highlighting the gender gap in mental health outcomes in adolescents worldwide, with girls reporting worse average mental health than boys [52]. Although non-binary adolescents could not be included in the analyses due to their small sample size, our preliminary results suggest higher COVID-19-related distress and lower level well-being.

### 4.4. Limitations, Strengths, and Future Recommendations

The results of this study must be interpreted in light of its limitations. First, although we had measures before and during the pandemic, no causal link can be made—i.e., the changes observed between waves cannot be solely attributed to the effect of the pandemic. Moreover, we used only two time points; therefore, it was impossible to assess trajectories of changes. Thus, future studies including several time points are needed to better understand changes in mental health and psychological adaptation of adolescents over time.

All the measures used were validated, but not necessarily among the population that constituted the current sample of French—and majority Caucasian—adolescents. In addition, the low internal reliability of the externalizing behavior scale may have underestimated the changes over time. As we could not examine the hypothesized associations among non-binary adolescents, future studies should be conducted with a larger sample of this unique and important group. Furthermore, we had a moderate attrition rate between baseline and follow-up, which is also a problem shared by several longitudinal studies during the pandemic. Attrition analyses revealed differences: participants who dropped out at follow-up reported higher levels of symptoms at T1. However, because the analyses used FIML to handle missing data, which can accurately estimate the coefficients of participants with missing data points [45], the risk of bias was significantly diminished. Nonetheless, caution should be taken when generalizing the results. Finally, it was impossible to measure ongoing CM during the pandemic in the present study. Considering that the pandemic has increased the proportion of youth who experienced CM due to confinement at home [53], it may also have been a factor that exacerbates youth mental health concerns. Future studies should therefore include various and validated measures of well-being to obtain a more complete picture of adolescents and to continue monitoring CM during major events, such as the COVID-19 pandemic. A better understanding of why adolescents with a history of CM experienced pandemic-related distress differently could also be explored in future studies.

Beyond these measurement issues and sample biases, important strengths of this research are the use of a large sample size and the longitudinal design, including the measurement of outcomes before the pandemic, combined with robust and appropriate statistical techniques.

## 5. Conclusions

The present study supports the existing literature concerning the negative effects of the COVID-19 pandemic on the well-being of adolescents. It also expands knowledge by highlighting two factors that should be considered to better understand to what degree the pandemic has affected the mental health and psychological adaptation of adolescents, namely, antecedents of CM, and the extent of distress experienced in relation to the pandemic. Our results also suggest that a year after the beginning of the pandemic, adolescents who reported moderate to high levels of COVID-19-related distress struggled more than their peers who had lower levels of distress. Thus, targeted interventions aimed at preventing deterioration in the psychological state of adolescents are necessary to counter the effects of the ongoing health crisis. Moreover, among the participants who reported antecedents of CM, although they appeared to have experienced slight improvements during the pandemic, they also presented with lower levels of well-being than their counterparts. In fact, CM was associated with more distress, which in turn led to higher levels of externalized and internalized behaviors and to lower levels of self-esteem and life satisfaction. These results suggest an immediate need to mitigate the impacts posed by COVID-19-related distress on the mental health and safety of vulnerable adolescents.

## Figures and Tables

**Table 1 ijerph-19-10172-t001:** Descriptive statistics of study variables.

		Girls	Boys	Nonbinary	Total Sample		
Outcome	Range	*M* (*SD*)	*M* (*SD*)	*M* (*SD*)	*M* (*SD*)	Skew	Kurtosis
Child maltreatment	0–5	1.19 (1.32)	0.87 (1.15)	2.42 (1.86)	1.13 (1.32)	1.08	0.26
COVID-19-related distress	0–80	31.1 (15.07)	18.3 (12.62)	44.2 (19.47)	25.11 (15.44)	0.6	−0.04
Internalizing behaviors T1	0–10	4.23 (2.91)	2.12 (2.41)	5.75 (2.99)	3.05 (2.83)	0.77	−0.41
Internalizing behaviors T2	0–10	4.15 (2.91)	1.78 (2.26)	4.2 (2.68)	3.04 (2.89)	0.78	−0.37
Externalizing behaviors T1	0–8	1.29 (1.3)	1.29 (1.39)	3.08 (2.32)	1.4 (1.42)	1.62	3.77
Externalizing behaviors T2	0–7	1.23 (1.08)	1.07 (1.18)	1.6 (1.52)	1.17 (1.15)	1.26	1.98
Self-esteem T1	0–16	11.01 (3.4)	12.7 (3.16)	7.96 (4.25)	11.77 (3.44)	−0.79	0.46
Self-esteem T2	0–16	10.98 (3.49)	12.15 (3.28)	6.4 (4.1)	11.5 (3.47)	−0.88	0.93
Life satisfaction T1	0–30	21.25 (6.58)	22.94 (5.83)	14.35 (6.48)	21.41 (6.31)	−0.92	0.37
Life satisfaction T2	0–30	20.04 (6.41)	21.07 (6.41)	10.40 (5.77)	20.43 (6.45)	−0.81	0.11

COVID-19, coronavirus disease 2019; T1, pre-pandemic; T2, during the pandemic.

**Table 2 ijerph-19-10172-t002:** Correlations among the study variables.

Outcome	1	2	3	4	5	6	7	8	9
1. Child maltreatment	–								
2. COVID-19-related distress	0.31 ***	–							
3. Internalizing behaviors T1	0.32 ***	0.49 ***	–						
4. Internalizing behaviors T2	0.20 ***	0.65 ***	0.59 ***	–					
5. Externalizing behaviors T1	0.31 ***	0.14 ***	0.38 ***	0.14 ***	–				
6. Externalizing behaviors T2	0.24 ***	0.31 ***	0.23 ***	0.31 ***	0.42 ***	–			
7. Self-esteem T1	−0.32 ***	−0.35 ***	−0.45 ***	−0.38 ***	−0.19 ***	−0.19 ***	–		
8. Self-esteem T2	−0.16 ***	−0.36 ***	−0.31 ***	−0.42 ***	−0.08 ***	−0.21 ***	0.52 ***	–	
9. Life satisfaction T1	−0.39 ***	−0.32 ***	−0.45 ***	−0.31 ***	−0.28 ***	−0.22 ***	0.58 ***	0.38 ***	–
10. Life satisfaction T2	−0.29 ***	−0.34 ***	−0.32 ***	−0.37 ***	−0.15 ***	−0.24 ***	0.40 ***	0.61 ***	0.55 ***

Note: *** *p* < 0.001.

**Table 3 ijerph-19-10172-t003:** Results of repeated measures *t*-tests for the study variables.

Outcome	*t* (*df*)	*p*	*d*
Internalizing behaviors	1.71 (773)	0.087	0.06
Externalizing behaviors	2.29 (774)	0.023	0.08
Self-esteem	2.56 (776)	0.011	0.09
Life satisfaction	7.13 (767)	<0.001	0.26

**Table 4 ijerph-19-10172-t004:** Results of moderation analyses: factors that moderate changes from T1 to T2.

Outcome	Child Maltreatment				COVID-19-Related Distress				Gender			
	*b*	S.E.	*p*	*β*	*b*	S.E.	*p*	*β*	*b*	S.E.	*p*	*β*
Internalizing behaviors	–0.33	0.07	<0.001	–0.16	0.04	0.01	<0.001	0.22	–0.22	0.18	0.220	–0.05
Externalizing behaviors	–0.16	0.04	0.001	–0.07	0.01	0.004	0.001	0.18	0.03	0.10	0.785	–0.01
Self-esteem	0.42	0.09	<0.001	0.16	–0.02	0.01	0.014	–0.09	0.60	0.24	0.014	0.09
Life satisfaction	0.46	0.17	0.006	0.10	–0.04	0.02	0.009	–0.09	1.04	0.45	0.020	0.08

**Table 5 ijerph-19-10172-t005:** Results of the mediation models.

	**COVID-19-Related Distress**				**Internalizing Behaviors T2**			
**Model 1**	** *b* **	**S.E.**	** *p* **	** *β* **	** *b* **	**S.E.**	** *p* **	** *β* **
Child maltreatment	1.34	0.39	0.003	0.10	−0.05	0.06	0.420	−0.02
Internalizing behaviors T1	2.48	0.19	<0.001	0.46	0.36	0.04	<0.001	0.35
COVID-19-related distress					0.09	0.01	<0.001	0.49
*R^2^*	24.5%				51.6%			
Indirect effect	*β = 0*.05, 95% CI [0.02, 0.08]							
	**COVID-19-Related Distress**				**Externalizing Behaviors T2**			
**Model 2**	** *b* **	**S.E.**	** *p* **	** *β* **	** *b* **	**S.E.**	** *p* **	** *β* **
Child maltreatment	2.54	0.43	<0.001	0.22	0.08	0.03	0.011	0.09
Externalizing behaviors T1	0.98	0.44	0.027	0.09	0.31	0.04	<0.001	0.38
COVID-19-related distress					0.02	0.003	<0.001	0.23
*R^2^*	6.7%				26.2%			
Indirect effect	*β = 0*.05, 95% CI [0.03, 0.07]							
	**COVID-19-Related Distress**				**Self-Esteem T2**			
**Model 3**	** *b* **	**S.E.**	** *p* **	** *β* **	** *b* **	**S.E.**	** *p* **	** *β* **
Child maltreatment	1.66	0.41	<0.001	0.14	0.08	0.09	0.382	0.03
Self-esteem T1	−1.41	0.17	<0.001	−0.31	0.47	0.04	<0.001	0.46
COVID-19-related distress					−0.05	0.01	<0.001	−0.21
*R^2^*	14.6%				31.6%			
Indirect effect	*β =* −0.03, 95% CI [−0.05, −0.02]							
	**COVID-19-Related Distress**				**Life Satisfaction T2**			
**Model 4**	** *b* **	**S.E.**	** *p* **	** *β* **	** *b* **	**S.E.**	** *p* **	** *β* **
Child maltreatment	1.61	0.43	<0.001	0.14	−0.35	0.20	0.076	−0.07
Life satisfaction T1	−0.63	0.10	<0.001	−0.26	0.48	0.05	<0.001	0.47
COVID-19-related distress					−0.08	0.01	<0.001	−0.19
*R^2^*	11.2%				34.4%			
Indirect effect	*β =* −0.03, 95% CI [−0.05, −0.01]							

Note: Indirect effects were obtained through COVID-19-related distress.

## Data Availability

The data presented in this study are available on request from the corresponding author. The data are not publicly available due to ethical restrictions, which were used under license for the current study and are therefore not publicly available.

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
