# Peer review of "Changes in Canadian Adolescent Well-Being since the COVID-19 Pandemic: The Role of Prior Child Maltreatment"

_ijerph, 2022, doi:10.3390/ijerph191610172_

Round 1

Reviewer 1 Report

Thank you for the opportunity to review this interesting manuscript. I am making the following suggestions to improve the reader's experience and increase their understanding of your work.

- There are 6 key terms used throughout the paper but none have been defined. As your results and conclusions are tied directly to these terms and they are used often, I highly recommend that they be defined.

Child maltreatment - you do note the items that the students could endorse on your measure - I will leave it up to you if you think that is enough

Internalizing and Externalizing  - these in my opinion are not generally understood and should be a priority to define.

Well-being is noted first and then the language changes to life satisfaction and self-esteem (based on the measures chosen) - I would suggest a sentence that defines well-being in those terms.

I would also suggest a Recommendations section for future research - this would be helpful to further the work that you have completed. 

Limitations - I am familiar with some of the measures that you have used - given the population of students and their self-disclosed cultural backgrounds it would be appropriate and important to note when a measure used has not been normed for the population using it. Most measures have only been normed using a caucasian, western, english speaking population and one must be careful when using them with other populations. 

Reviewer 2 Report

Thank you for giving me the opportunity to read and comment a report “Changes in Canadian adolescent well-being since the COVID-19 pandemic: The role of prior child maltreatment”, by Dion J, et al.

In the reviewed manuscript, the changes in Canadian adolescent well-being, from before the onset of the pandemic to 1 year after the pandemic began has been investigated.

This paper is well written, correctly structured with a suitable research concept, the study limitations are addressed, and it is of relevance to readers of the journal. However, I include a comments for your consideration.

·       In the abstract, it would be appropriate to standardize the acronym COVID-19, since it appears in different forms: COVID-19 and COVID.

·       The "Results" section should not contain opinions of the authors, just describe the findings aseptically. For example, expressions like “As expected…(line 280), should be removed.

·       Please delete section "6.Patents" as it does not apply to this manuscript.
